# Denoising Non-Stationary Signals via Dynamic Multivariate Complex Wavelet Thresholding

**DOI:** 10.3390/e25111546

**Published:** 2023-11-16

**Authors:** Kim C. Raath, Katherine B. Ensor, Alena Crivello, David W. Scott

**Affiliations:** 1Imperative Global; kcraath@gmail.com; 2Department of Statistics, Rice University, MS-138, Houston, TX 77005, USA; ensor@rice.edu; 3Chevron Co., San Ramon, CA 94583, USA; ixmocane@gmail.com

**Keywords:** continuous wavelet transform, data-driven and adaptive thresholding, partial density estimation, integrated squared error, *WaveL*
_2_
*E*, nonparametric method

## Abstract

Over the past few years, we have seen an increased need to analyze the dynamically changing behaviors of economic and financial time series. These needs have led to significant demand for methods that denoise non-stationary time series across time and for specific investment horizons (scales) and localized windows (blocks) of time. Wavelets have long been known to decompose non-stationary time series into their different components or scale pieces. Recent methods satisfying this demand first decompose the non-stationary time series using wavelet techniques and then apply a thresholding method to separate and capture the signal and noise components of the series. Traditionally, wavelet thresholding methods rely on the discrete wavelet transform (DWT), which is a static thresholding technique that may not capture the time series of the estimated variance in the additive noise process. We introduce a novel continuous wavelet transform (CWT) dynamically optimized multivariate thresholding method ($WaveL_{2}E$). Applying this method, we are simultaneously able to separate and capture the signal and noise components while estimating the dynamic noise variance. Our method shows improved results when compared to well-known methods, especially for high-frequency signal-rich time series, typically observed in finance.

## 1. Introduction

Various time series in economics and finance are formed by non-stationary processes. For instance, Exchange Traded Funds (ETF’s) consist of combinations of different equities functioning at different periodicities. In order to understand and identify dynamic associations, it is necessary to explore the characteristics within each individual time series and the changing dynamics across a series. Over the past decade, there has been a significant increase in the use of time-varying spectral representations to analyze dynamic time series behavior in finance. Recently, emphasis has been placed on using wavelets. Rostan and Rostan [1] mention that quantitative analysts (quants) “nibbling” with wavelets are heavily being recruited by major hedge funds. They specifically point out that the interest in wavelet analysis has spiked in recent years because new methods and techniques have come about to analyze physical phenomena, such as seismic and electrical signals, that propagate through time in waveform. These methods are being adapted and applied to economic and financial time series as these signals also propagate through time in waveform. The application of time-varying techniques to augment traditional portfolio management tools by distinguishing across multiple investment horizons or scales is becoming a growing field of interest as well [2,3,4]. Chaudhuri and Lo [5], who coined the term *spectral portfolio theory*, suggest that the flexibility of the wavelet transform could be used to overcome various difficulties of the Fourier transform for spectral portfolio analysis. Wavelet transforms, when specified correctly, are simple to interpret and can add tremendous value to the quantitative finance models that financial engineers implement. Wavelets have also proven useful in characterizing two-dimensional fractional Brownian fields, often used in financial studies [6]. For an expanded discussion of wavelet theory and implementation examples in finance, see [7].

The decomposition of time series into their component pieces is frequently applied, but further inference such as optimal dynamic *thresholding* and *forecasting* has only increased in interest over the past few years. A recent paper by Reményi and Vidakovic [8] proposes adaptive wavelet denoising methodology using a fully Bayesian hierarchical model in the complex wavelet domain. They compare their results to two methods proposed by [9]. These authors also compare their methods to the well-known Empirical Bayesian Threshold (EBayesThresh) of [10] and various other superior universal hard and soft thresholding methods such as the SureShrink methods of [11]. Even though these two papers were written ten years apart and are over a decade old, they are still commonly referenced and can be seen as the gold standard in the domain of wavelet thresholding. For a survey of other wavelet denoising or thresholding techniques and their comparisons, see [12]. More recent thresholding (denoising) methods were developed by the authors of [9] who proposed two new complex-valued wavelet techniques, while the authors of [13] proposed a data-driven threshold called the SureBlock, which uses Stein’s unbiased risk estimate (SURE) criterion. He et al. [14] designed a new threshold, considering interscale correlation by improving the basic universal hard threshold in order to address the issue of removing too many wavelet coefficients. Each method improved characteristics that are lacking in the existing “gold standard” methods.

In general, hard thresholding reduces all coefficients to zero that do not exceed the threshold. Soft thresholding pushes toward zero any coefficient whose magnitude exceeds the threshold, and zeros the coefficient otherwise. As a comparison to our proposed method, we specifically use the SureShrink methods for the hard and soft thresholds. We also compare our methods to the EBayesThresh. We choose these methods for our comparison as they show superior results to the more recent and well-cited methods pointed out earlier in this paper.

The choice of wavelet family is very important. Most solutions are not based on the continuous wavelet transform (CWT) due to the difficulty of having a tractable solution or an analytical solution [1]. The CWT is calculated by shifting, continuously, a scalable function over a signal and by calculating the correlation between the function and signal, resulting in potential redundancies. However, we mitigate these redundancies as well as address both the tractability and analytical issues by uniquely choosing the Morlet wavelet with the exact specifications for concise interpretation, see Section 2 and [15] for more details.

The novel wavelet thresholding method we propose in this paper builds upon work on minimum distance estimation, initiated by the authors of [16,17]. We propose decomposing each univariate time series into scale-specific wavelet coefficients using a CWT. After decomposition, we apply Scott’s multivariate minimum distance partial density estimation ($L_{2}E$) to each time increment. In doing so, we separate each time point into a mixture model of signal and noise without having to explicitly assume a model for the signal. Upon analyzing these results, we are not only able to find the optimal signal, but we also estimate the variance of the noise component. In essence, we model the wavelet coefficients as a two-component mixture, where the signal component is unspecified and the noise component is Gaussian; see the Methods section for more details. The $L_{2}E$ (distance criterion) can be used to exclude groups of outliers; therefore, by estimating the partial density parameters of the noise component, we do not have to make assumptions on the signal component. Ultimately, we are able to distinguish between signal and noise, by taking our signal to be “outliers” while assuming that the noise has a known structure which we can model.

Section 2 describes the $WaveL_{2}E$ method, starting with a mathematical exposition of wavelets in Section 2.1. We use [18,19,20] as examples to explain wavelet characteristics and introduce statistical methods that extend the capabilities of wavelets to visualize unexpected structures in multidimensional data in Section 2.1.1. We then introduce the $WaveL_{2}E$ in Section 2.2 and further identify how our method compares to other well-established wavelet thresholding methods, using simulated time series that have dynamically changing behaviors. In Section 3, we discuss and display the comparative analysis. In this section, we also note that our $WaveL_{2}E$ results yield improved outcomes with added information and interpretations. Before concluding in Section 5 with the strengths of the new methods and a list of the next potential steps and applications, we introduce an example in Section 4. The selected example addresses the investment relationships in water and energy ETF’s [19]. In this example, we highlight the post-thresholding improved coherence analysis of the water–energy nexus.

## 2. Methods

### 2.1. Wavelets

Wavelets are mathematical functions that decompose temporally localized frequency components, providing information of each component with a resolution matched to its scale. Wavelet analysis in economics and finance has demonstrated promising results. This has led to a resurgent of the literature focused on wavelet transforms and denoising.

#### 2.1.1. CWT

A wavelet is usually derived by scaling and shifting a mother wavelet $\psi \left(t\right)\in L^{2}\left(R\right)$ into daughter wavelets $\psi_{a,b}\left(t\right)\in L^{2}\left(R\right)$:(1)$$\psi_{a,b}\left(t\right)=\frac{1}{\sqrt{a}}\psi \left(\frac{t-b}{a}\right),$$
where a>0 defines the scale and b∈R defines the shift. Given a time series $x\left(t\right)\in L^{2}\left(R\right)$, the continuous wavelet transform (CWT) provides a decomposition of x(t) into time-scale components:(2)$$W_{x}\left(a,b\right)=\langle x,\psi_{a,b}\rangle =\int_{R}x\left(t\right)\psi_{a,b}^{*}\left(t\right)\;dt,$$
where ∗ denotes the complex conjugate. Usually the CWT coefficients are presented in a power spectrum, where wavelet power is defined as $|W_{x}{\left(a,b\right)|}^{2}$ and visualized in the time-scale {b,a} half plane with logarithmic scale *a*-axis (vertical) increasing upward and a linear time-scale *b*-axis (horizontal) (see Figure 1). More generally, the CWT can be seen as a set of continuous band-pass filters applied to a time series.

Formally, ψ(t) is a mother wavelet if the following three conditions are met: (1) $\int_{R}\psi \left(t\right)dt=0$, (2) $\int_{R}\psi^{2}\left(t\right)dt=1$, and (3) it satisfies an admissibility condition. The admissibility condition ensures the reconstruction of a time series from its wavelet transform:(3)$$x\left(t\right)=\frac{1}{a^{2}C_{\psi }}\int_{0}^{\infty }\left[\int_{R}W_{x}\left(a,b\right)\psi_{a,b}\left(t\right)db\right]da,$$
where $C_{\psi }=\int_{R}\frac{{\left|\hat{\psi }\left(\omega \right)\right|}^{2}}{\left|\omega \right|}d\omega$ and $\hat{\psi }\left(\omega \right)$ is the Fourier transform of ψ(t) in the CWT. By the convolution theorem, the CWT, $W_{x}\left(a,b\right)$, of a discrete time series x(t) can be approximated by averaging *n* (the length of the time series) convolutions for each scale using the discrete Fourier transform (DFT) of x(t) and $\hat{\psi }\left(\omega \right)$ [21]. Traditionally, the constant ($C_{\psi }$) is called the wavelet admissible constant and a wavelet whose admissible constant satisfies $0<C_{\psi }<\infty$ is called an admissible wavelet. The most commonly used mother wavelet in finance and economic research is the Morlet wavelet:(4)$$\Psi^{M}\left(t\right)=\pi^{-\frac{1}{4}}e^{-\frac{1}{2}t^{2}}e^{i\omega_{0}t},$$
where $e^{i\omega_{0}t}$ is the complex sinusoid, $e^{-\frac{1}{2}t^{2}}$ is a Gaussian envelope with a standard deviation of one, and $\pi^{-\frac{1}{4}}$ normalizes the wavelet to have unit energy, i.e., satisfying the admissibility conditions. As suggested in [22,23], we set the central frequency $\omega_{0}=6$. This value provides a reasonable trade-off between frequency localization and temporal evolution. Choosing a Morlet wavelet with the above specifications provides us with unique properties described in [15]. First, by choosing a complex wavelet, we have a separation of information between the phase and amplitude. This allows us to examine leading–lagging relationships depicted in Section 4 (see [19] for more details). Furthermore, our method is able to recover the original series without using the inverse transform. However, the Morlet wavelet is also analytic, allowing us to recover the original time series through an inverse transform. Second, ref. [24] describes three different ways to convert scales into frequency. These special frequencies are all equal to the central frequency, $\omega_{0}$, for the Morlet wavelet. Using the usual definition of the relation between scale and Fourier frequency, defined as $f\left(s\right)=\frac{\omega_{0}}{2\pi s}$, as well as selecting $\omega_{0}=6$, results in $f\left(s\right)\approx \frac{1}{s}$, which provides better interpretability. Further, with this parameterization, the standard deviation in both time and frequency are equal, proving an optimal relationship between time and frequency accuracy. The current literature indicates that the choice of Morlet wavelet with $\omega_{0}=6$ proves to be best when using wavelets for feature extraction purposes in economic research. Once again, for a further breakdown of the above summarized specifications, see [15]. A good source for different wavelets and their inherent characteristics can be found in [25].

### 2.2. $WaveL_{2}E$ Threshold

In this section, we combine the CWT and the $L_{2}E$ method. Scott [16] proposed minimizing the Integrated Square Error (ISE) with the aim of minimizing the squared distance between two probability density curves using the $L_{2}$ estimation criterion (referred to as the $L_{2}E$). In Appendix A, we show that for a general multivariate mixture distribution:(5)h(x)=wf(x|θ)+(1−w)g(x),
where g(x) is an unspecified density, and w∈(0,1) is a mixing parameter, then the $L_{2}E$ for θ=(w,σ) is found by (numerically) minimizing
(6)$$L_{2}E\left(w,\sigma \right)=\frac{w^{2}}{{\left(2\sigma \sqrt{\pi }\right)}^{d}}-\frac{2w}{n}\sum_{i=1}^{n}\phi \left(x_{i}|0,\sigma^{2}I_{d}\right).$$

To formulate the dynamic selection of the coefficients of the multivariate continuous wavelet transform using the $L_{2}E$, we first describe our time series y with the simple signal plus noise model:(7)y=q(t)+ϵ,
where $y=(y_{1},\ldots ,y_{n})$ denotes the observations at our *n* time points, the signal is defined by $q\left(t\right)=(q\left(t_{1}\right),\ldots ,q\left(t_{n}\right))$, $q\left(t\right)\in L^{2}\left(R\right)$, and $\epsilon =\left(\epsilon_{1},\ldots ,\epsilon_{n}\right)\overset{iid}{∼}N\left(0,\sigma^{2}I_{n}\right)$, with unknown variance $\sigma^{2}$. Taking the continuous wavelet transform (W) of the time series, we can further express the additive noise model as:(8)Wy=Wq(t)+Wϵ.
Framing the additive noise model in Equation (Equation 8) as a general multivariate mixture distribution, we equate Wy to h(x) of Equation (Equation 5), with the strong signal components of the wavelet coefficients, Wq(t), assigned to (1−w)g(x). The remaining wavelet coefficients are categorized as noise, and are combined with Wϵ, which is equated to wf(x|θ) of Equation (Equation 5). With this specification, minimizing (Equation 6) allows us to estimate the noise component parameters without making assumptions of the number or distribution of the ‘significant’ wavelet coefficients embodied in g(x). Since we assume i.i.d Gaussian noise with variance $\sigma^{2}$, we also assume that the pure noise wavelet coefficients are i.i.d with covariance matrix $\sigma^{2}I_{d}$. Thus, the wavelet coefficients with the covariance matrix, $\sigma^{2}I_{d}$, are assumed to be independent. This assumption of independence means that we only have two parameters to estimate, namely σ and *w*, regardless of the dimension.

After minimizing the criteria and recovering our parameters $\left(\hat{\sigma },\hat{w}\right)$, we apply two thresholding methods to the wavelet coefficients based on these estimates. We apply the first of these thresholds to the distribution of the squared wavelet coefficients, referred to as the $WaveL_{2}E$. The reason we apply the minimization to the squared wavelet coefficients is due to our assumptions that we have standard Gaussian noise, the square of the wavelet coefficients representing the Gaussian noise is therefore assumed to be distributed $\chi^{2}$. The second threshold is determined using the 95%$\chi^{2}$ critical value of the the squared wavelet coefficients distribution. This threshold is referred to as the $WaveL_{2}E_{\chi^{2}}$. The squared wavelet coefficients are also the power of the signal; in Section 2, we present these in a CWT power spectrum. After applying the threshold, the power spectrum can be used to visualize the applied thresholds. For a simple simulated demonstration of these thresholding methods, see Figure 2. This figure visualizes an additive noise model, with three stationary signals, before and after thresholding, using both the $WaveL_{2}E$ and the $WaveL_{2}E_{\chi^{2}}$ thresholding methods. We are able to recover the pure signal for the three periods in the original additive noise model by removing the induced noise across all wavelet coefficients. These methods work well for signals that are constant or stationary. However, when there are inter-scale dependencies, the static version of these thresholds do not necessarily extract all the signals. Therefore, we introduce a dynamic version of the static $L_{2}E$ criterion.

### 2.3. Dynamic $WaveL_{2}E$ Threshold

Minimum distance estimators, like the $L_{2}E$, identify the portion of the data that matches the model and separates the data that do not match. Scott [16] showed that the inefficiencies (asymptotically) of the parameters estimated using the maximum likelihood estimator versus the minimization of the ISE or the $L_{2}E$ criterion are roughly that of the the mean versus the median in a general statistical analysis. The $L_{2}E$ criterion specified in [16] for mixture distributions is therefore very good at identifying groups of outliers in large datasets. This minimum distance estimator is also robust, without requiring additional specifications (meta-parameters) that are needed in other likelihood estimators, making it an ideal candidate for thresholding.

In order to evaluate the changing weight, $w_{t}$, and the changing noise variance, $\sigma_{t}$, across time, the implementation of the $L_{2}E$ minimization needs to be adapted to optimally estimate $\left({\hat{w}}_{t},{\hat{\sigma }}_{t}\right)$ at each time-localized observation.

Therefore, by adapting Equation (Equation 6) for time-dependency, we change the $L_{2}E$ criterion to obtain unique estimates ${\hat{\sigma }}_{t}$ and ${\hat{w}}_{t}$. The result is the dynamic optimization by minimizing the following $L_{2}E$ criterion:(9)$$L_{2}E\left(w_{t},\sigma_{t}\right)=\frac{w_{t}^{2}}{{\left(2\sigma_{t}\sqrt{\pi }\right)}^{d}}-\frac{2w_{t}}{n}\sum_{i=1}^{n}\phi \left(x_{i}|0,\sigma_{t}^{2}I_{d}\right).$$

We apply the same two thresholding methods mentioned previously, based on these estimates. The only difference in the adjusted implementation is that we apply these thresholds to the distribution of the squared wavelet coefficients at each time point. Figure 3 is a simple demonstration of the steps followed to implement the $WaveL_{2}E$ method and the resulting estimates recovered. The steps for implementation are as follows (see our R Package referenced in Data Availability Statement for more detailed implementation instructions):Calculate the wavelet transform of the observed time series and recover the squared wavelet coefficients.Apply the $WaveL_{2}E$ and $WaveL_{2}E_{\chi^{2}}$ at each time point, solving for the estimates $\left({\hat{w}}_{t},{\hat{\sigma }}_{t}\right)$ of the $L_{2}E$ criterion.Threshold the wavelet coefficients based on the estimates.Calculate the inverse transform of the denoised time series or solve for the pure signal using the recovered estimates.

Further extending and generalizing Equation (Equation 9), we include intra-scale dependence (within scale) by evaluation blocks (windows) of time across scales. We perform this by enabling block sizes of length *h*. The first block is labeled as block zero, with estimates $\left({\hat{w}}_{0},{\hat{\sigma }}_{0}\right)$. The $WaveL_{2}E$ and $WaveL_{2}E_{\chi^{2}}$ are applied to each of these blocks, across the scales; see Figure 3. For example, if we identify a time window (block) of 22 days, an average fiscal month, we would have estimates for $\left({\hat{w}}_{t},{\hat{\sigma }}_{t}\right)$, at t=h, where (h=0,22,44,66,…,T), with *T* being the length of the series. When h=T, we have the static $L_{2}E$ criterion described in Equation (Equation 6). More specifically, the generalization of the $L_{2}E$ criterion then becomes:(10)$$L_{2}E\left(w_{h},\sigma_{h}\right)=\frac{w_{h}^{2}}{{\left(2\sigma_{h}\sqrt{\pi }\right)}^{d}}-\frac{2w_{h}}{n}\sum_{i=1}^{n}\phi \left(x_{i}|0,\sigma_{h}^{2}I_{d}\right).$$

## 3. Results and Comparative Analysis

### 3.1. Inter-Scale and Intra-Scale

In this section, our main goal is to show comparative results of our methods against the EBayesthresh and SureShrink referenced in Section 1. We refer to these methods as the gold standard for wavelet thresholding. To further understand our findings, we examine the percentage of significance area (PSA) and the percentage of total volume (PTV) given in [27]. We use these two measures to estimate the statistically significant area maintained after our threshold implementation and the portion of volume in the CWT power spectrum, respectively. Thus, to simplify, PSA measures the percentage of the wavelet coefficients removed by our method and PTV measures the resulting statistical significance captured by our method.

Current thresholding methods have two limiting assumptions. First, most methods assume wavelet coefficients, representative of the signal components, which follow a specified parametric model. Unlike Bayesian methods with hyperparameters that need to be tuned, our multivariate wavelet thresholding method is adaptive and data-driven, so no specifications are needed. Second, traditional wavelet thresholding methods assume independence of all wavelet coefficients. However, wavelet correlation theory suggests that signal coefficients are not independent and only the wavelet coefficients of noise are uncorrelated (see [28] for more details). These methods generally assume that the smallest level of wavelet coefficients are likely to contain pure noise. Both SureShrink and EbayesThresh methods use adaptions of the median absolute deviation (MAD) of the finest level of coefficients to estimate the noise variance of the signal. However, these MAD estimates of the smallest scale wavelet coefficients overestimate the variance in high-frequency data when compared to our noise variance estimate. Overestimation results in too large a threshold. Visualizations of these comparative results for the different estimates of the noise variance can be found in [17]. Therefore, noise variance has to be estimated in a data-driven, dynamic manner as we suggest in our proposed wavelet thresholding method, the $WaveL_{2}E$.

The assumption that all levels of wavelet coefficients are independent leads to thresholding methods that do not capture both the inter-scale (between) and intra-scale (within) dependencies. Figure 4 summarizes these differences by pointing out how the $WaveL_{2}E$ thresholding method mitigates these within and between scale dependencies. Figure 3 provides a simple demonstration of how the $WaveL_{2}E$ method incorporates between scale dependencies, by evaluating the additive noise model at each increment, and within scale dependency, by allowing blocks of time to be evaluated simultaneously. Our method has the flexibility to separate the noise and signal components by minimizing the $L_{2}E$ criterion across scales and for specific time windows. In the next subsection, we will show how this flexibility results in better, simplified thresholding and the interpretation of commonly simulated signals. This subsection then leads to an example of how our method can be implemented to analyze real- world applications.

### 3.2. Comparative Analysis

We start our comparative analysis by analyzing the root-mean-squared error (RMSE) of both the gold standard methods and our method for the simulated periodic time series in Figure 2. Figure 5 extends the previous static analysis and includes the dynamic analysis from Equation (Equation 9) for both the $WaveL_{2}E$ and $WaveL_{2}E_{\chi^{2}}$ thresholds. The static implementation removed a significant amount of the coefficients, whereas the dynamic implementation was more selective. This property becomes more relevant when the signals are not as simple as what was simulated in this example. In Table 1, we point out the percentage total volume (PTV) and the percentage significance area (PSA). These two measures refer to the percentage of the total wavelet coefficients that represent the estimated true signal after threshold implementation and the percentage of the total statistical significance represented by these thresholded wavelet coefficients. The static implementation thresholds about 80% of the coefficients for the $WaveL_{2}E$ and about 92% of the coefficients for the $WaveL_{2}E_{\chi^{2}}$. The remaining wavelet coefficients represent about 94% and about 55% of the statistical significance for the respective thresholding methods. On the other hand, the dynamic implementation thresholded about 42% and about 50% of the wavelet coefficients for the respective methods, but this implementation was able to maintain about 99% of the statistical significant wavelet coefficients for both methods.

In Table 2, we include the root-mean-squared error (RMSE) for the wavelet SureShrink (WavShrink) universal hard threshold (RMSE: 0.1120), the WavShrink universal soft threshold (RMSE: 0.2205), and the empirical Bayes threshold (RMSE: 0.0734). Our static implementation for the $WaveL_{2}E$ threshold has an RMSE of 0.0775 and for the $WaveL_{2}E_{\chi^{2}}$ threshold has an RMSE of 0.1271. The generalization and implementation of the dynamic method results in an RMSE of 0.0931 and an RMSE of 0.0963 for the $WaveL_{2}E$ and $WaveL_{2}E_{\chi^{2}}$ thresholds, respectively.

As we pointed out earlier in this section and in Figure 4, most traditional methods assume independence between scales (inter-scale). When we only have stationary signals, as we see in Figure 5, the EBayesThresh method slightly outperforms the new method we introduce in this paper. However, the moment high-frequency data and inter-scale dependency are introduced, see Figure 6 and Table 3, our method outperforms the other methods when implementing the static $L_{2}E$ criterion, and the results are equivalent in RMSE when applying the dynamic $L_{2}E$ criterion. We also see similar results for the PTV and PSA in Table 4, where the dynamic implementation maintains more of the statistical significant wavelet coefficients. The signal we identified to analyze with the aforementioned specifications consists of a ten-year cycle (as the first periodic component) and the second periodic component is a one-year cycle that changes to a three-year cycle between 10 and 20 years and a five-year cycle between 50 and 60 years with variance that changes five times, i.e., every 20 years. The simulated series represents 100 years of monthly values, or 1200 observations. Specifically,
(11)$$y_{t}=cos\left(\frac{2\pi t}{p_{1}12}\right)+cos\left(\frac{2\pi t}{p_{2}12}\right)+\epsilon_{t},\;t=\left(1,2,\ldots ,1200\right),$$
where $\epsilon_{t}∼N\left(0,p_{3}\right)$ with $p_{3}=\left(0.5,0.05,0.25,0.15,0.05\right)$ with $p_{1}=10$, $p_{2}=3$ when $10\leq \frac{t}{12}\leq 20$, $p_{2}=5$ when $50\leq \frac{t}{12}\leq 60$, and $p_{2}=1$ otherwise. Figure 6 shows two different analyses, the first using the static $L_{2}E$ criterion and the second using the dynamic $L_{2}E$ criterion.

Introducing more inter-scale dependency, we analyze eight different signals by implementing the dynamic $L_{2}E$ criterion. We also add to the analysis the empirical $WaveL_{2}E$ calculation ($EWaveL_{2}E$). After recovering both $\left({\hat{w}}_{t},{\hat{\sigma }}_{t}\right)$ and since we are assuming independence, we can recover the denoised signal from the original equation by solving for the estimated true signal, $\hat{g}\left(x\right)$:(12)$$\hat{g}\left(x\right)=\frac{h\left(x\right)-{\hat{w}}_{h}\phi \left(x|0,{\hat{\sigma }}_{h}^{2}I_{d}\right)}{\left(1-{\hat{w}}_{h}\right)}.$$

Equation (Equation 12) provides us with the flexibility to create a potential forecasting model; however, this task is left for future research. In Table 5, the last column contains the RMSE of an adjusted $EWaveL_{2}E$ for all the signals. We made an adjustment to ensure that we can include the results of various different signals. We use the median absolute deviation (MAD) of each of the estimates $\left({\hat{\sigma }}_{h},{\hat{w}}_{h}\right)$ to recover the estimated $\hat{g}\left(x\right)$. The time series and CWT power spectrum of each of these signals are visualized in Figure 7. These spectra clearly show the inter-scale dependencies.

Table 5 and Table 6 are the RMSE results of the eight different signals with two different signal-to-noise ratio’s (SNR), 2 and 5, respectively. As soon as there is more inter-scale dependency and the signals are not only noise, our model outperforms. Another interesting observation is the difference between the empirical calculation and the inverse wavelet transform. It seems that the inversion does yield better results. However, the adjusted empirical results ($EWaveL_{2}E$) compared to the $WaveL_{2}E$ and $WaveL_{2}E_{\chi^{2}}$ for some of the signals are very similar. To show the value in our new method, we expand the example of the water–energy nexus in the next section (see [19]).

## 4. Motivating Example: Water–Energy Nexus

Raath and Ensor [19] quantify the dynamic relationship between energy and water commodities by applying wavelet techniques to better understand the dynamic relationship that exists in the water–energy nexus. Using daily water and energy commodity ETF price data from 2007 to 2017, they evaluate the respective wavelet transforms of each time series. We use this water–energy nexus quantification as a motivating example in explaining each component of our novel thresholding method.

After thresholding both the water (CGW ETF) and energy (XLE ETF) prices and returns, as seen in Figure 1, we further analyze by comparing the wavelet-squared coherence (WSC) before and after thresholding, using the $WaveL_{2}E$. The WSC analyzes local linear correlations in regions of statistical significant co-movement and can be used to define investment horizon-specific behavior. In Figure 8, the top two plots represent the before (left) and after (right) thresholding results. It can clearly be seen that there are less statistically significant co-moving components in the WSC plot after thresholding (top right). We also note that during the financial crisis, from 2008 to 2012, there were no statistically significant behaviors at the weekly and biweekly investment horizons (scale); the relationship was only noise.

The next set of plots point out the partial wavelet coherence conditional on the market (SPY ETF); for more details, see [15]. We wanted to eliminate the market effects on the water–energy nexus and then evaluate the leading–lagging (bottom right) relationship for the quarterly investment horizon (bottom left and center). The quarterly investment horizon was identified because of the statistically significant co-movement during the 2014–2016 oil glut (top right). The phase difference (bottom left) of our thresholded series does not seem to have any significant leading–lagging relationship. The light purple and darker green indicate that water is leading and the darker purple and lighter green indicate that energy is leading (bottom right). However, as soon as we implement the partial wavelet coherence on our thresholded series, we see significant leading–lagging relationships. Raath and Ensor [19] pointed out the significance of water-leading energy during the 2014–2016 oil glut, which is very clearly confirmed here using our $WaveL_{2}E$ thresholding method.

We point out in Figure 9 the plots of the estimates for $\left({\hat{\sigma }}_{h},{\hat{w}}_{h}\right)$, where h=22, a fiscal month (top) on the XLE ETF. We added these here to show the flexibility of our method and to visually demonstrate that a static weight or variance would not work for the water–energy nexus.

In this figure, we also add the reconstructed price series and return series using our $WaveL_{2}E$ threshold method. In the return plot (bottom right), we also include the EWaveL2E estimate of $\hat{g}\left(x\right)$ from Equation (Equation 12) using only the h=22 estimates $\left({\hat{\sigma }}_{h},{\hat{w}}_{h}\right)$.

## 5. Conclusions

In this paper, we present a method that denoises non-stationary time series by dynamic multivariate complex wavelet thresholding. We demonstrate that our method performs best among a broad class of methods for the type of series seen in finance. Our method performs especially well in environments when there are inter-scale dependencies. Further, we are able to capture not only the signal, but also an estimate of the variance for the additive noise of a time series. Other objectives met in this manuscript include: (1) choosing a continuous wavelet transform to increase interpretability while incorporating and adjusting the $L_{2}E$ criterion to optimize a data-driven threshold; (2) increasing the dynamic adaptability by generalizing our $WaveL_{2}E$ method; (3) describing how to implement these thresholds in practice; and (4) developing a user-friendly R package, which implements the thresholds introduced in this paper.

Our methodology is applied to a financial series, addressing the water–energy nexus [19]. By denoising the price series, using our methodology, we form a deeper understanding of the relationship between the price series representing water and that representing energy. These results would be difficult to discern without implementing our threshold. We confirmed that the water–energy nexus depends on general market behavior. When the market behavior is removed, we find that water prices lead energy prices at the quarterly investment horizon during the 2014–2016 oil glut.

An added feature of our thresholding method is the estimation of the variance of the additive noise at specific localized windows of time. These estimates can be used to forecast the volatility series, work which will be further explored in future papers. Future investigations also include revisiting wavelet estimation for temporal random fields, as in [6].

## Figures and Tables

**Figure 1 entropy-25-01546-f001:**
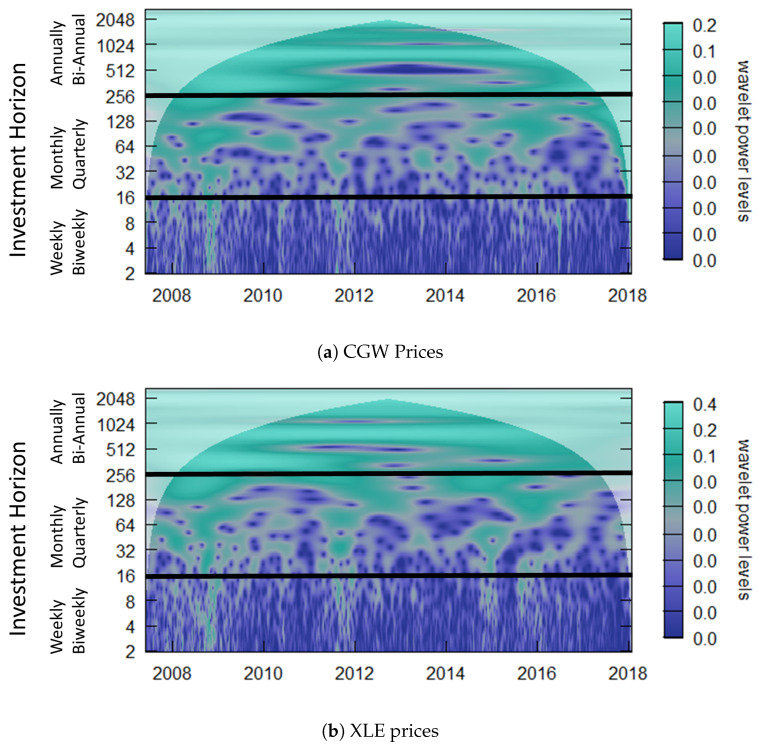
The continuous wavelet transform (CWT) power spectrum of water (**a**,**c**) and energy (**b**,**d**) commodity ETF prices (**a**,**b**) and returns (**c**,**d**). These plots are visual representations of the power spectrum of each individual series. The investment horizons (vertical axis) are such that the value one through fourteen represents weekly and biweekly investment horizons. Sixty-four days represent a quarterly investment horizon, whereas 250 and above represent annual and larger investment horizons. The horizontal axis indicates the 10 years of data. The white overlay defines the cone of influence.

**Figure 2 entropy-25-01546-f002:**
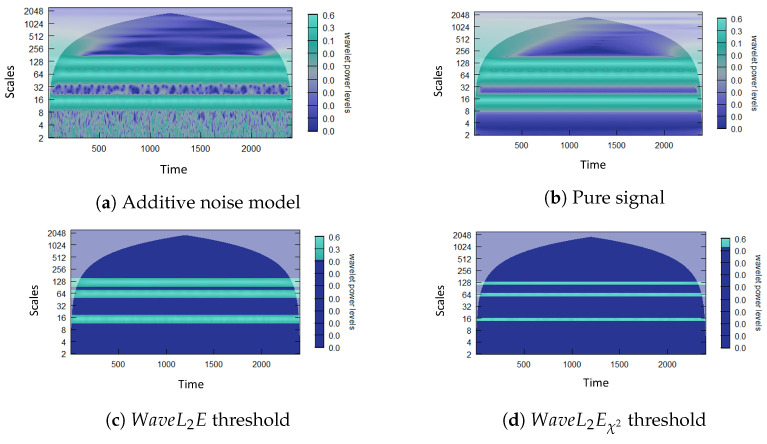
The CWT power spectrum (**a**) of a simple additive noise model with three stationary signals. Biweekly (15 days), quarterly (64 days), and biannually (125 days). Hence, we have that: $y_{t}=sin\left(\frac{2\pi t}{15}\right)+sin\left(\frac{2\pi t}{64}\right)+sin\left(\frac{2\pi t}{125}\right)+\epsilon$, where ϵ∼N(0,0.1) and t=(1,…,2400). Also, this figure shows the CWT power spectrum (**b**), which is the pure signal without noise, the CWT power spectrum (**c**) after the $WaveL_{2}E$ threshold, and the CWT power spectrum (**d**) after the $WaveL_{2}E_{\chi^{2}}$ threshold. For the minimization process, we use constrained optimization using PORT routines with 0≤σ<∞ and 0≤w<1 [26].

**Figure 3 entropy-25-01546-f003:**
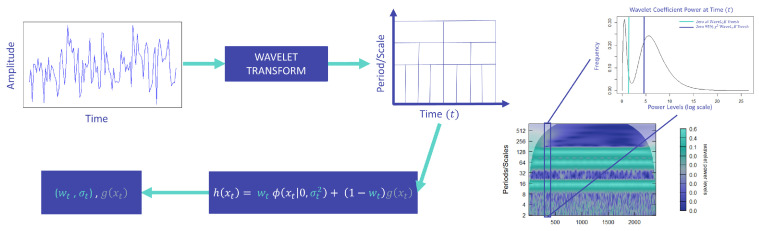
Demonstration of the process applied to a time series. Starting with the observed time series, we apply the Morlet wavelet, a continuous wavelet transform (CWT), and then after decomposition, we minimize the $L_{2}E$ criterion. This minimization provides us with the optimal estimates for $\sigma_{t}$ and $w_{t}$. After estimation, we can remove the noise component and recover the signal component of the original observed series.

**Figure 4 entropy-25-01546-f004:**
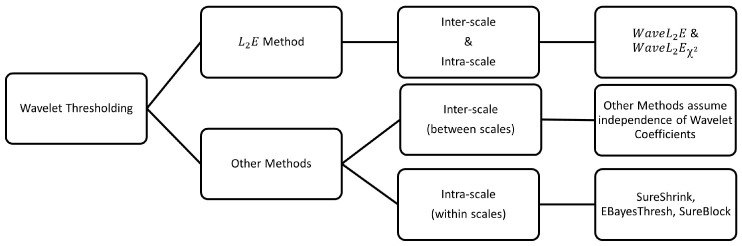
Summary of the difference between the wavelet $L_{2}E$ thresholding method we introduce in this paper and the other gold standard methods. The fundamental difference is that our novel method incorporates both inter-scale and intra-scale dependencies with no limitations on the estimates of the noise variance or parametric specifications.

**Figure 5 entropy-25-01546-f005:**
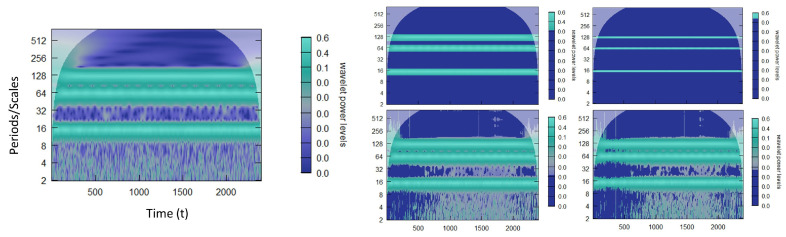
Two different comparative analyses: The first plot (**left**) is the base CWT power spectrum for the signal $y_{t}=sin\left(\frac{2\pi t}{15}\right)+sin\left(\frac{2\pi t}{64}\right)+sin\left(\frac{2\pi t}{125}\right)+\epsilon_{t}$, where $\epsilon_{t}∼N\left(0,0.1\right)$ and t=(1,…,2400). The top analyses is the $WaveL_{2}E$ (**left**) and $WaveL_{2}E_{\chi^{2}}$ (**right**) thresholds from Equation (Equation 6) and the bottom analysis is the dynamic $WaveL_{2}E$ (**left**) and $WaveL_{2}E_{\chi^{2}}$ (**right**) thresholds from Equation (Equation 9).

**Figure 6 entropy-25-01546-f006:**
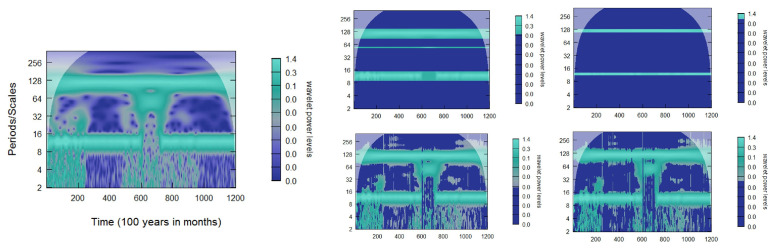
Two different comparative analyses. The first plot (**left**) is the base CWT power spectrum for the signal from Equation (Equation 11). The top analyses is the $WaveL_{2}E$ and $WaveL_{2}E_{\chi^{2}}$ thresholds from Equation (Equation 6) and the bottom analysis is the dynamic $WaveL_{2}E$ and $WaveL_{2}E_{\chi^{2}}$ thresholds from Equation (Equation 9).

**Figure 7 entropy-25-01546-f007:**
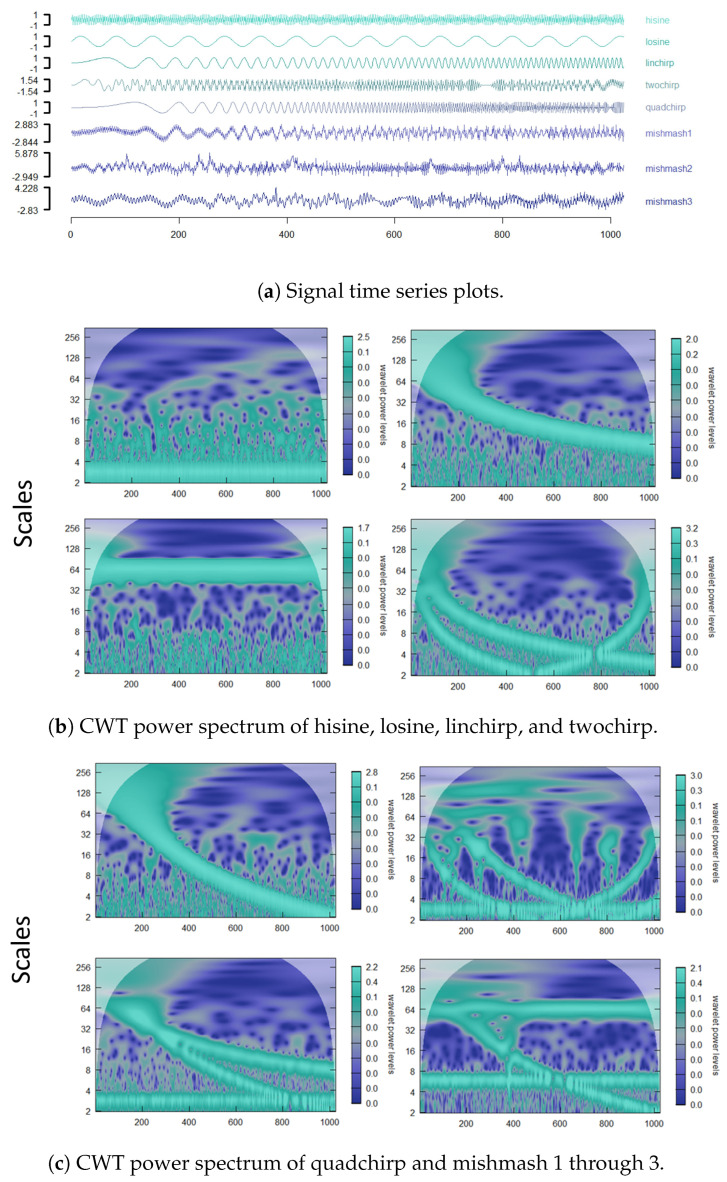
Comparative analysis of eight traditional time series depicted in the top panel (7a) before a wavelet transform. The series are labeled hisine, losine, linchirp, twochirp, quadchirp, mishmash1, mishmash2 and mismash3. The CWT for hisine, losine, linchirp and two chirp is given in panel 7b reading from left to right and top to bottom, whereas the CWT for the remaining four series is provided in panel 7c, namely quadchirp and mishmash1 through mishmash3.

**Figure 8 entropy-25-01546-f008:**
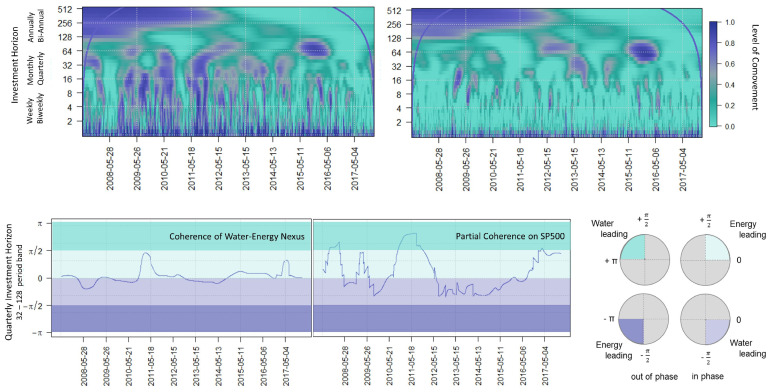
Analyzing the wavelet-squared coherence (WSC) of the water–energy nexus before (**top left**) and after (**top right**) $WaveL_{2}E$ thresholding. We evaluate both the WSC (**bottom left**) and the partial coherence (**bottom middle**) results of the quarterly investment horizon by analyzing the leading–lagging relationship (**bottom right**) of the nexus.

**Figure 9 entropy-25-01546-f009:**
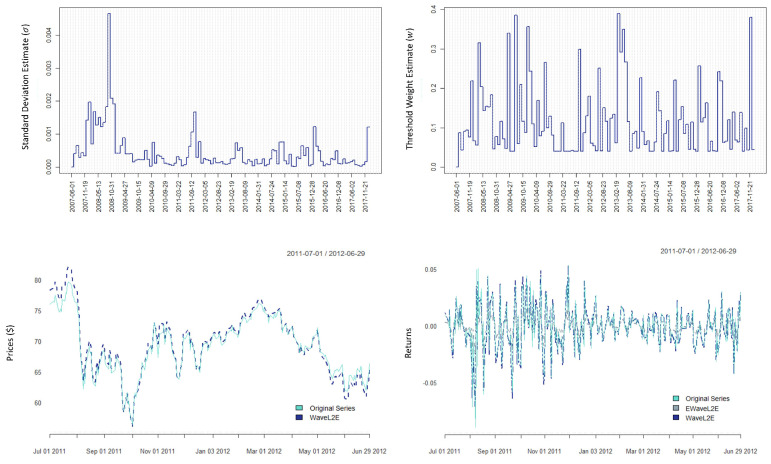
Analyzing and recovering the fiscal month, on the 22nd day; estimates for the $WaveL_{2}E$ and visualizing the results of both the $WaveL_{2}E$ and the $EWaveL_{2}E$ after inversion and recovery of the estimated true time series. Method executed on the XLE ETF.

**Table 1 entropy-25-01546-t001:** The percentage total volume (PTV) and percentage significance area (PSA) of the static and dynamic $WaveL_{2}E$ and $WaveL_{2}E_{\chi^{2}}$ thresholding methods, for the simulated series defined in Figure 2.

Analysis	${\mathrm{WaveL}}_{2}E$	${\mathrm{WaveL}}_{2}E_{\chi^{2}}$
Static PTV	20.48%	7.80%
Static PSA	94.10%	54.59%
Dynamic PTV	57.77%	50.79%
Dynamic PSA	99.84%	99.64%

**Table 2 entropy-25-01546-t002:** RMSE comparison of our method using the static $L_{2}E$ criterion and the dynamic $L_{2}E$ criterion to the gold standard (GS) methods. The RMSE of the WaveShrink (WShrink) hard (H) and WShrink soft (S) thresholds as well as the EBayes threshold, for the simulated series defined in Figure 2.

Analysis	WShrink (H)	WShrink (S)	EBayes	${\mathrm{WaveL}}_{2}E$	${\mathrm{WaveL}}_{2}E_{\chi^{2}}$
Static RMSE	0.1120	0.2205	0.0734	0.0775	0.1271
Dynamic RMSE	0.1120	0.2205	0.0734	0.0931	0.0963

**Table 3 entropy-25-01546-t003:** RMSE comparison of our method using the static $L_{2}E$ criterion and the dynamic $L_{2}E$ criterion to the gold standard (GS) methods. The RMSE of the WaveShrink (WShrink) hard (H) and WShrink soft (S) thresholds as well as the EBayes threshold, for the simulated series from model (Equation 11).

Analysis	WShrink (H)	WShrink (S)	EBayes	${\mathrm{WaveL}}_{2}E$	${\mathrm{WaveL}}_{2}E_{\chi^{2}}$
Static RMSE	0.2740	0.2939	0.2430	0.1844	0.2682
Dynamic RMSE	0.2740	0.2939	0.2430	0.2535	0.2547

**Table 4 entropy-25-01546-t004:** The percentage total volume (PTV) and percentage significance area (PSA) of the static and dynamic $WaveL_{2}E$ and $WaveL_{2}E_{\chi^{2}}$ thresholding methods, for the simulated series from model Equation 11.

Analysis	${\mathrm{WaveL}}_{2}E$	${\mathrm{WaveL}}_{2}E_{\chi^{2}}$
Static PTV	20.54%	7.03%
Static PSA	87.44%	55.81%
Dynamic PTV	75.17%	70.81%
Dynamic PSA	99.93%	99.86%

**Table 5 entropy-25-01546-t005:** Using a signal-to-noise ratio (SNR) of two, we analyze the RMSE comparison of our method using the dynamic $L_{2}E$ criterion to the gold standard (GS) methods for eight different signals: the RMSE of the WaveShrink (WShrink) hard (H) and WShrink soft (S) thresholds, as well as the EBayes threshold. We identify the smallest RMSE with a star (*).

Signal	WShrink (H)	WShrink (S)	EBayes	${\mathrm{WaveL}}_{2}E$	${\mathrm{WaveL}}_{2}E_{\chi^{2}}$	${\mathrm{EWaveL}}_{2}E$
hisine	0.7069	0.7069	0.7116	0.3443	0.3425 *	0.3832
losine	0.1457	0.2813	0.1301 *	0.3196	0.3201	0.3642
linchirp	0.3145	0.4520	0.2203 *	0.3296	0.3257	0.4026
twochirp	0.9662	0.9728	0.9315	0.4897	0.4835 *	0.5543
quadchirp	0.5694	0.6232	0.4812	0.3416 *	0.3543	0.4018
mishmash1	1.1166	1.1635	1.0778	0.6493	0.6478 *	0.7096
mishmash2	1.3406	1.3620	1.2559	0.8034 *	0.9486	0.9420
mishmash3	0.9354	1.0594	0.6603	0.5919	0.5899 *	0.6702

**Table 6 entropy-25-01546-t006:** Using a signal-to-noise ratio (SNR) of five, we analyze the RMSE comparison of our method using the dynamic $L_{2}E$ criterion to the gold standard (GS) methods for eight different signals: the RMSE of the WaveShrink (WShrink) hard (H) and WShrink soft (S) thresholds, as well as the EBayes threshold. We identify the smallest RMSE with a star (*).

Signal	WShrink (H)	WShrink (S)	EBayes	${\mathrm{WaveL}}_{2}E$	${\mathrm{WaveL}}_{2}E_{\chi^{2}}$	${\mathrm{EWaveL}}_{2}E$
hisine	0.7068	0.7068	0.7076	0.1371	0.1359 *	0.1619
losine	0.0810	0.1339	0.0559 *	0.1248	0.1207	0.1830
linchirp	0.1378	0.2498	0.0925 *	0.1295	0.1929	0.1804
twochirp	0.9668	0.9727	0.9127	0.2145 *	0.2389	0.2901
quadchirp	0.3507	0.5088	0.1744	0.1541 *	0.2338	0.2791
mishmash1	1.1156	1.1637	1.0772	0.2693 *	0.3241	0.3078
mishmash2	1.3386	1.3618	1.2360	0.5202 *	0.6562	0.5630
mishmash3	0.7803	0.9463	0.4679	0.2525 *	0.2567	0.3467

## Data Availability

R-package for WaveL2E routine: The R-package CoFESWave contains code used to perform the diagnostic methods described in the article. Data and software used in this article may be obtained from https://github.com/kcraath/CoFESWaveResearch (accessed on 9 November 2023) and https://github.com/kcraath/CoFESWave (accessed on 9 November 2023), respectively.

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
