# Peer review of "Denoising Non-Stationary Signals via Dynamic Multivariate Complex Wavelet Thresholding"

_entropy, 2023, doi:10.3390/e25111546_

Round 1
Reviewer 1 Report
Comments and Suggestions for Authors
Report on ‘Denoising Non-stationary Signals by Dynamic Multivariate Complex Wavelet Thresholding’ by Kim C Raath et al.
The paper presents a novel method, WaveL2E, for denoising non-stationary time series through dynamic multivariate complex wavelet thresholding. It decomposes time series into scale-specific wavelet coefficients using a continuous wavelet transform and a specific multivariate minimum distance partial density estimation (L2E). The choice of the distance criterion helps in determining the outliers. The presented method, in some cases, outperforms standard techniques. Authors use the context of financial time series to show the advantage of their method, which, in short, lies in the ability to capture the inter-scale dependencies.
I would like to point out only a few minor issues
- line 162: “an the”
- section 2.2 is somehow challenging to follow; maybe an appendix with more detailed calculations would help with that
- almost all Figures overlap captions
- the majority of plots are hard to read; please increase the font size used on plots
The paper emphasises increasing dynamic adaptability by generalising the WaveL2E method and shows how to implement these thresholds in practice. Also, developing an R routine (CoFESWave) is helpful. The package mentioned in section 2.3 and in section 6 is hard to find. The work mentions the Supplementary Materials, but as the reviewer, I did not have access to these materials, and, in turn, I cannot state the usefulness of the actual code.
The practical implementation and accuracy of the method need further investigation. Nonetheless, the work is precise, well written and deserves publication in Entropy, which I recommend.
Reviewer 2 Report
Comments and Suggestions for Authors
Review of "Denoising Non-stationary Signals by Dynamic Multivariate Complex Wavelet Thresholding" by Raath et al. (2023)
Authors presented a novel method based on wavelets to denoising non-stationary signals in multivariate complex time series. Simulation and applications illustrate the performance of the method. I think that paper has merit to be published in Entropy journal. However, some technical difficulties and typos errors were detected during my review. Thus I recommend to authors to address these issues and other suggestions, before I could recommend publication of the manuscript:
1. L34: in "([2], [3], and [4])", authors could add the reference: "Nicolis, O., Mateu, J., Contreras-Reyes, J.E. (2020). Wavelet-based entropy measures to characterize two-dimensional fractional Brownian fields. Entropy, 22(2), 196", because increment the literature about wavelet estimation in random fields.
2. L126: $\omega_0$ is the frequency? if yes, please add it.
3. Eq. (7): this equation only presents an integral. I recommend to add something like $ISE(f,h)=\int ...$. Also, add the support of the integral.
4. L158: $\hat{\int}$ is a weird notation. Thus I recommend $\hat{ISE}(f,h)$. Then fix the equations of (8). Also, it is not clear what is the estimator of the integral. It is related to Riemmann integral?
5. Eq. (10): $w$ could be confused with frequency of Fourier transform. Use another letter such as $\alpha$. Also add that $\alpha\in[0,1]$.
6. After Eq. (12): you mean gaussian densities about the result in [25]?
7. Eq. (17): 100 <-> 1 ?
8. L399: Also, another further work could be wavelet estimation for temporal random fields (Nicolis et al., 2020).
Typos errors:
1. L14: remove "Supplementary materials for your article are available online" (this could not be mentioned in abstract).
2. L34: "Chaudhuri [5], who...".
3. L42: "A recent paper by" <-> "Reményi and Vidakovic".
4. L53: "He et al. [13] designed".
5. L72: "we develop and describe" <-> "proposed".
6. L81: "normal (Gaussian)" <-> "Gaussian".
7. L162: "the" appears as strikethrough.
8. L228-232: Put these lines with normal font size.
9. Table 2 (and other ones): It is not necessary to put these number in red because you described these results in lines 298-304.
10. L343 & 368: "Raath and Ensor [18]".
Round 2
Reviewer 2 Report
Comments and Suggestions for Authors
All my previous comments have been well addressed by the authors. I don't have further comments.